# Differential plasmacytoid dendritic cell phenotype and type I Interferon response in asymptomatic and severe COVID-19 infection

**Martina Severa**[1⊚], **Roberta A. Diotti**[2⊚], **Marilena P. Etna**[1⊚], **Fabiana Rizzo**[1], **Stefano Fiore**[1], **Daniela Ricci**[1], **Marco Iannetta**[3], **Alessandro Sinigaglia**[4], **Alessandra Lodi**[3], **Nicasio Mancini**[2], **Elena Criscuolo**[2], **Massimo Clementi**[2], **Massimo Andreoni**[3], **Stefano Balducci**[5], **Luisa Barzon**[4], **Paola Stefanelli**[1], **Nicola Clementi**[2‡], **Eliana M. Coccia**[1‡*]

**1** Department of Infectious Diseases, Istituto Superiore di Sanità, Rome, Italy, **2** Laboratory of Medical Microbiology and Virology, Vita-Salute San Raffaele University, Milan, Italy, **3** Infectious Disease Clinic, Policlinico Tor Vergata, Rome, Italy, **4** Department of Molecular Medicine, University of Padova, Padua, Italy, **5** Metabolic Fitness Association, Monterotondo, Rome, Italy

⊚ These authors contributed equally to this work.
‡ These authors are joint senior authors on this work.
* eliana.coccia@iss.it

**Data Availability Statement:** All relevant data are within the manuscript and its Supporting Information files.

## Abstract

SARS-CoV-2 fine-tunes the interferon (IFN)-induced antiviral responses, which play a key role in preventing coronavirus disease 2019 (COVID-19) progression. Indeed, critically ill patients show an impaired type I IFN response accompanied by elevated inflammatory cytokine and chemokine levels, responsible for cell and tissue damage and associated multi-organ failure. Here, the early interaction between SARS-CoV-2 and immune cells was investigated by interrogating an *in vitro* human peripheral blood mononuclear cell (PBMC)-based experimental model. We found that, even in absence of a productive viral replication, the virus mediates a vigorous TLR7/8-dependent production of both type I and III IFNs and inflammatory cytokines and chemokines, known to contribute to the cytokine storm observed in COVID-19. Interestingly, we observed how virus-induced type I IFN secreted by PBMC enhances anti-viral response in infected lung epithelial cells, thus, inhibiting viral replication. This type I IFN was released by plasmacytoid dendritic cells (pDC) *via* an ACE-2-indipendent but Neuropilin-1-dependent mechanism. Viral sensing regulates pDC phenotype by inducing cell surface expression of PD-L1 marker, a feature of type I IFN producing cells. Coherently to what observed *in vitro*, asymptomatic SARS-CoV-2 infected subjects displayed a similar pDC phenotype associated to a very high serum type I IFN level and induction of anti-viral IFN-stimulated genes in PBMC. Conversely, hospitalized patients with severe COVID-19 display very low frequency of circulating pDC with an inflammatory phenotype and high levels of chemokines and pro-inflammatory cytokines in serum. This study further shed light on the early events resulting from the interaction between SARS-CoV-2 and immune cells occurring *in vitro* and confirmed *ex vivo*. These observations can improve our understanding on the contribution of pDC/type I IFN axis in the regulation of the anti-viral state in asymptomatic and severe COVID-19 patients.

**Funding:** This work was financially supported by Istituto Superiore di Sanità (www.iss.it) and partly co-financed by the Italian Ministry of Health (grant GR-2016-02363749 to MS) and the European Union's Horizon 2020 research and innovation programme, under grant agreement no. 874735 (VEO) to LB. The funders had no role in study design, data collection and analysis, decision to publish, or preparation of the manuscript and none of the authors received a salary from any of the funders.

## Author summary

SARS-CoV-2 pandemic has resulted in millions of infections and deaths worldwide, yet the role of host innate immune responses in COVID-19 pathogenesis remains only partially characterized. Innate immunity represents the first line of host defense against viruses. Upon viral recognition, the secretion of type I and III interferons (IFN) establishes the cellular state of viral resistance, and contributes to induce the specific adaptive immune responses. Moving from *in vitro* evidences on the protective role played by plasmacytoid dendritic cells (pDC)-released type I IFN in the early phase of SARS-CoV-2 infection, here we characterized *ex vivo* the pDC phenotype and the balance between antiviral and pro-inflammatory cytokines of COVID-19 patients stratified according to disease severity. Our study confirms in COVID-19 the crucial and protective role of pDC/type I IFN axis, whose deeper understanding may contribute to the development of novel pharmacological strategies and/or host-directed therapies aimed at boosting pDC response since the early phases of SARS-CoV-2 infection.

## Introduction

As of middle of August, 2021, COVID-19 pandemic, caused by severe acute respiratory syndrome coronavirus 2 (SARS-CoV-2), has resulted in more than two hundred and seven million cases and more than four million deaths globally (https://covid19.who.int/).

Five to ten% of SARS-CoV-2 infected subjects progress, unfortunately, to severe or critical disease, requiring mechanical ventilation and admission to intensive care unit [1,2]. Host characteristics, including age (immunosenescence) and comorbidities (hypertension, diabetes mellitus, lung and heart diseases) may influence the course of the disease [3]. Severe pneumonia caused by SARS-CoV-2 is marked by immune system dysfunction and systemic hyperinflammation leading to acute respiratory distress syndrome, macrophage activation, hypercytokinemia and coagulopathy [4].

Interestingly, in the majority of the cases SARS-CoV-2 infection can be asymptomatic with a rate of presentation estimated to be at least around 17% of total cases [5]. Understanding the immunological features of these subjects is challenging but is of key importance to comprehend early events controlling SARS-CoV-2 replication.

Progression and severity of SARS-CoV-2 infection are, indeed, strictly related to virus-triggered immune dysfunctions often associated to defects in type I Interferon (IFN) production, and to over-production of pro-inflammatory cytokines, strongly implicated in the resulting damage to the airways and to the organs [6]. Therefore, disease severity is not only due to the virus direct damage but strongly depends on the host inflammatory response to the infection [6–8].

For instance, type I IFNs emerged as key protective factors in preventing COVID-19 disease severity. Genetical analyses of COVID-19 patients with life-threatening pneumonia had, indeed, reported inborn errors of type I IFN immunity and variants in genes encoding for RNA sensors and type I IFN regulating elements in absence of other risk factors [9,10]. Furthermore, more than 10% of COVID-19 patients with severe pneumonia had neutralizing auto-antibodies against either type I IFN-$\varpi$ or IFN-$\alpha$s, or both, before disease onset and 94% of them were males; findings that might also explain the excess of men among the individuals with life-threatening COVID-19 disease [11]. Moreover, severe COVID-19 patients uniquely produce antibodies that functionally block, through binding of the Fc domain to Fc$\gamma$R-IIb, the production of type I IFN and ISG-expressing cells found in mild disease [12].

Plasmacytoid dendritic cells (pDC) are important mediators of antiviral immunity through their specific ability to produce high levels of type I IFN upon sensing of the majority of viruses and, to this aim, they developed a highly specialized intracellular machinery to recognize viral nucleic acids [13]. They were shown to sense and control, *via* type I IFN release, infection with both coronaviruses SARS-CoV and Middle East respiratory syndrome coronavirus (MERS) [14–16]. Importantly, pDC were found significantly reduced or absent in the blood of patients with severe COVID-19 [17–19] and were absent in the lungs [20] and in bronchoalveolar lavage fluids [21] while showed a higher proportions in moderate disease.

In this paper, we investigate the SARS-CoV-2-elicited innate immune response using a human PBMC-based *in vitro* experimental model. We evaluate the key role of pDC in the induction of type I IFN response in this context. Also, we performed an in-depth analysis of pDC phenotype and IFN and cytokine levels in COVID-19 patients, stratified according to disease severity, to evaluate possible correlation with COVID-19 progression and subsequent life-threatening complications.

## Results

### *In vitro* SARS-CoV-2 stimulation of human PBMC induces a TLR7/8-dependent cytokine and chemokine production

To understand if cells of immune system can directly sense SARS-CoV-2 leading to activation of innate response, we stimulated human PBMC with a clinical SARS-CoV-2 isolate (hCoV-19/Italy/UniSR1/2020).

To monitor the virus kinetic in human PBMC, we first treated these cells with SARS-CoV-2 at different multiplicity of infection (MOI; 0.01, 0.02, 0.04 and 0.1) and then viral titration was performed at the inoculum check at time zero (T0), as well as at 24 (T1) and 48 (T2) hours post-infection (**S1 Fig**). In spite of an unaltered cell viability (**S1 Table**), a sharp decrease was observed in viral titer. These results indicate that PBMC are not permissive, at least *in vitro*, to infection.

Having in mind that SARS-CoV-2 infection of both upper and lower airways regulates type I and III IFN responses [22], here we studied by real-time PCR the expression of these anti-viral cytokines in the mixed immune cell population of PBMC and found a dose-dependent induction of high levels of type I IFN-αs and IFN-β, as well as type III IFN-λ1 transcripts, as compared to cells stimulated with the TLR7/8 ligand R848, used as positive control (**Fig 1A**). In line with these data, we found a strong release of IFN-αs in culture supernatants (**Fig 1B**) and a concomitant induction of Mx dynamin-like GTPase 1 (Mx1) expression (**Fig 1C**), one of the classical IFN-inducible genes representing the so-called anti-viral IFN-signature. No change in the level of cytokines and gene expression was observed in mock-treated PBMC, conditioned with supernatants from uninfected Vero E6 (**Fig 1**).

Thus, in human PBMC SARS-CoV-2 was unable to interfere efficiently with the intracellular machinery responsible for anti-viral IFN release and induction of IFN signature. Moreover, the addition of a specific inhibitor of TLR7/8 signaling (I-TLR7/8) completely abolished SARS-CoV-2-induced type I IFN response in PBMC (**Fig 1**), suggesting that the viral RNA may be involved in induction of IFN production in this setting. Efficacy and specificity of TLR7/8 inhibition was proven in control experiments in which PBMC were cultured with TLR7, TLR7/8, TLR9 ligands as well as viral stimulations (SARS-CoV-2 and the TLR7-dependent Influenza A H1N1 virus) pre-treated with either the specific I-TLR7/8 or an unrelated ODN control (ODN ctrl) (**S2 Fig**). As expected, the addition of I-TLR7/8 reduced only the cytokine production induced by TLR7-dependent stimuli, while no change occurred in presence of ODN ctrl, proving a *bona-fide* effect of the inhibitor used in our experiments.

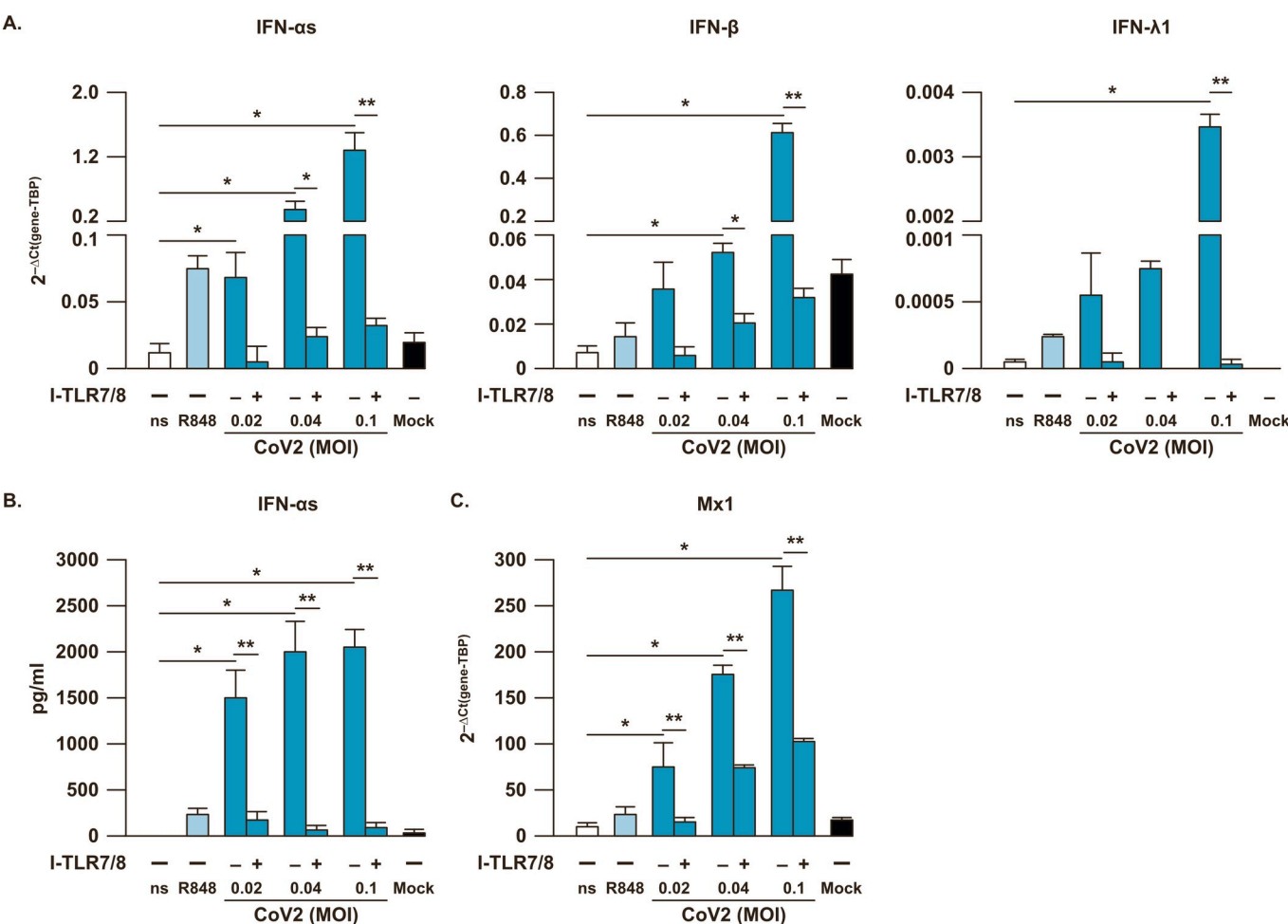

**Fig 1. SARS-CoV-2 stimulation induces a TLR7/8-dependent type I and III IFN response in PBMC.** Peripheral blood mononuclear cells (PBMC) were left untreated (not stimulated, ns) or Mock-treated as negative controls, or stimulated for 24 hours with the TLR7/8 agonist R848 (5μM) as positive control, with SARS-CoV-2 (CoV2) at different multiplicity of infection (MOI; 0.02, 0.04 and 0.1) in presence or absence of a 30 minute pre-treatment with a specific TLR7/8 inhibitor (I-TLR7/8, 1μM). (**A**) Relative expression of IFN-αs, IFN-β, IFN-λ1 genes was measured by quantitative real time PCR analysis. All quantification data were normalized to TBP level by using the equation $2^{-\Delta Ct}$. (**B**) Production of IFN-αs was tested by specific ELISA in 24 hour-collected cell culture supernatants. (**C**) Mx1 gene expression was quantified by real time PCR as described above. Shown results were mean relative values ± SEM of 5 independent experiments. P-values were depicted as follows: *p≤0.05; ** p≤0.001.

In addition, in the same experimental setting we found a remarkable production of inflammatory cytokines such as IL-6, TNF-α and IL-1β, which are well-recognized players of the cytokine storm occurring in COVID-19 patients [23], while IL-12p70 level was unchanged in SARS-CoV-2-stimulated PBMC (**Fig 2A**). Importantly, the anti-inflammatory IL-10 factor was also induced but its level was inversely related to virus MOI and to inflammatory cytokine release (**Fig 2A**). Therefore, as for IFNs, the production of pro-inflammatory cytokines is dependent on TLR7/8 signaling (**Fig 2A**).

The recruitment and coordination of specific subsets of leukocytes at the site of viral infection heavily relies on chemokine secretion. Thus, in supernatants of PBMC stimulated with increasing doses of SARS-CoV-2 we also tested the release of CXC motif chemokine ligand 10 (CXCL-10 or IFN-γ inducible protein, IP-10), CXCL-9 (or monokine induced by IFN-γ, MIG), C-C motif chemokine ligand 2 (CCL-2 or macrophage cationic peptide 1, MCP-1), CXCL-8 (or IL-8) and CCL-5 (or Rantes) (**Fig 2B**). Among the studied chemokines, CXCL-10,

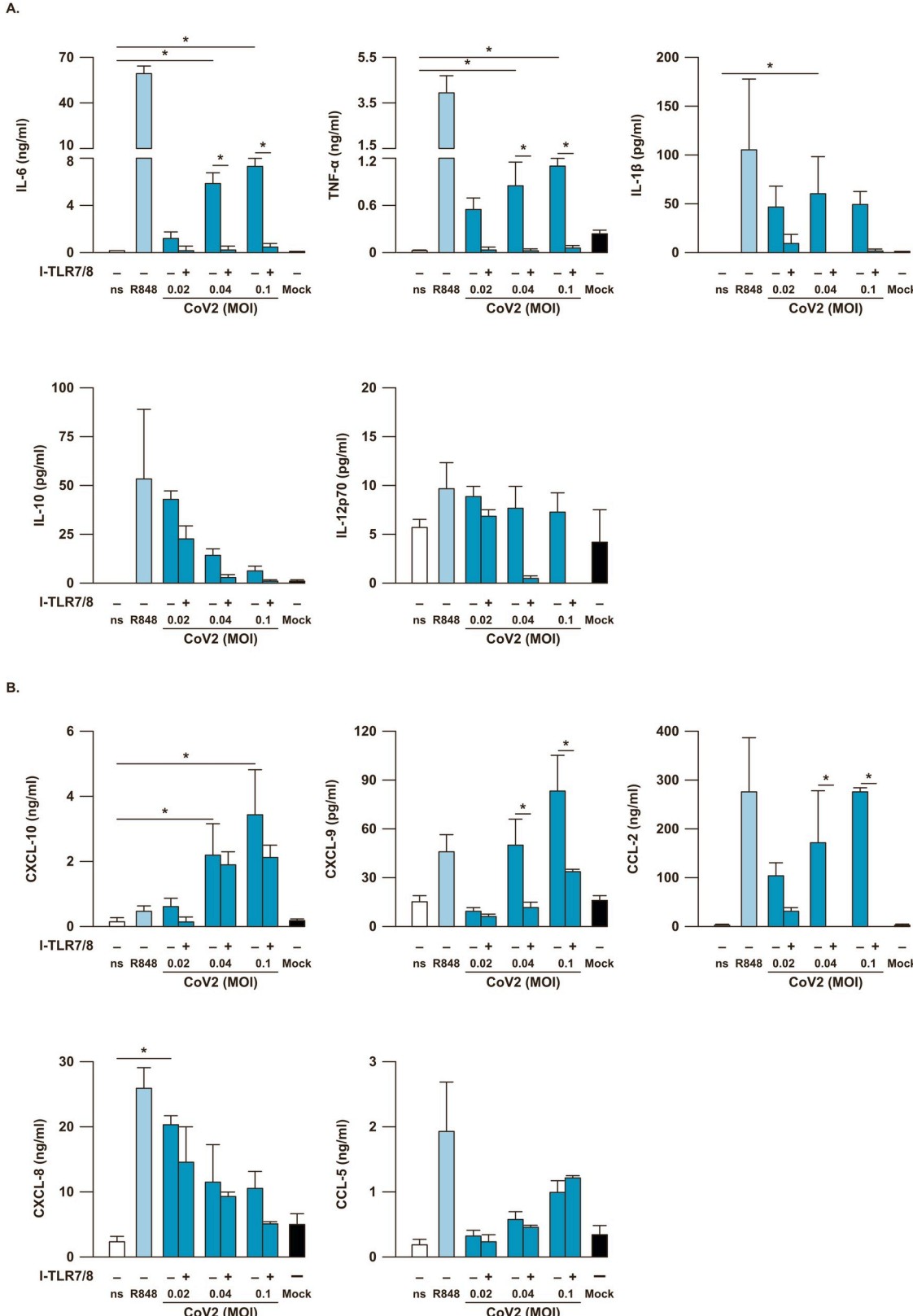

**Fig 2. SARS-CoV-2 stimulation induces a TLR7/8-dependent cytokine and chemokine production in PBMC.** Peripheral blood mononuclear cells (PBMC) were left untreated (not stimulated, ns) or Mock-treated as negative controls, or stimulated for 24 hours

with the TLR7/8 agonist R848 (5μM) as positive control, with SARS-CoV-2 (CoV2) at different multiplicity of infection (MOI; 0.02, 0.04 and 0.1) in presence or absence of a 30 minute pre-treatment with a specific TLR7/8 inhibitor (I-TLR7/8, 1μM). Production of cytokines (IL-6, TNF-α, IL-1β, IL-10, IL-12p70) (**A**) and chemokines (CXCL-10, CXCL-9, CCL-2, CXCL-8, CCL-5) (**B**) was tested by multiparametric cytokine bead arrays in collected cell culture supernatants. Shown results were mean relative values ± SEM of 5 independent experiments. P-values were depicted as follows: *p≤0.05.

CXCL-9 and CCL-2 were those with the highest induction in a viral dose-dependent manner (**Fig 2B**). In addition, our data showed that CXCL-9 and CCL-2 production was, in particular, TLR7/8-dependent (**Fig 2B**). The release of analyzed cytokines and chemokines was further increased at 48 hours post infection indicating an incremental SARS-CoV-2-dependent stimulation in our PBMC-based culture setting at the tested MOI (0.04 and 0.1) (**S3A and S3B Fig**).

## SARS-CoV-2-induced type I IFN secretion by human PBMC inhibits virus infection of Calu-3 cells

Several studies have clearly shown that SARS-CoV-2 infection modulates the release of type I and III IFNs [7,22,24]. In our study we observed that SARS-CoV-2 infection of the lung carcinoma epithelial cell line Calu-3 triggers IL-6 expression while was unable to induce Mx1 gene expression, suggestive of a strong reduction of type I IFN production (**Fig 3A**) and, thus, confirming what already observed [7,22].

Importantly, we first demonstrated that treatment of Calu-3 cells with supernatants collected from SARS-CoV-2-stimulated human PBMC (sup PBMC_CoV2) strongly induced the transcription of the IFN-inducible gene Mx1, as compared to cultures treated with supernatant from not stimulated PBMC (sup PBMC_ns) (**Fig 3A**). Then, to verify the biological activity of type I IFN naturally released by infected PBMC, we infected Calu-3 cells conditioned with either sup PBMC_ns or sup PBMC_CoV2 (**Fig 3B and 3C**). While the addition of sup PBMC_ns affected neither Mx1 transcription nor SARS-CoV-2 replication, the presence of sup PBMC_CoV2 correlated to an enhanced ISG signature and combined inhibition of SARS-CoV-2 infection as revealed by viral titrations of supernatants from Calu-3 cells (**Fig 3B and 3C**).

Hence, PBMC-derived type I IFN may rescue SARS-CoV2-mediated block of anti-viral response, thus, limiting viral expansion.

## Human pDC produce high levels of type I IFN upon SARS-CoV-2 stimulation

pDC are classified as the major type I IFN producing cells following viral infections by sensing viral RNAs *via* TLR7 [13]. Given the TLR7/8-dependent release of the anti-viral cytokines detected in SARS-CoV-2-stimulated PBMC, we hypothesized that pDC could be the main cell type responsible for type I IFN production. In addition, we also tested the possibility that the TLR8-expressing resting CD14$^+$ monocytes would be involved.

Thus, pDC were purified from PBMC of healthy donors and stimulated for 24 hours with SARS-CoV-2 (0.1 MOI) (**Fig 4**). In pDC cultures a strong production of IFN-αs, comparable to what found in PBMC, was detected, and this was clearly dependent on TLR7 signaling since I-TLR7/8 blocked release of this cytokine (**Fig 4A**). As mentioned above, efficacy and specificity of I-TLR7/8 treatment was also proven in control experiments in pDC in comparison to ODN ctrl (**S2B Fig**).

By using this setting, we also demonstrated that inactivation of SARS-CoV-2 by UV-C treatment [25,26] significantly reduces SARS-CoV-2-mediated IFN-α release in pDC (**Fig 4A**).

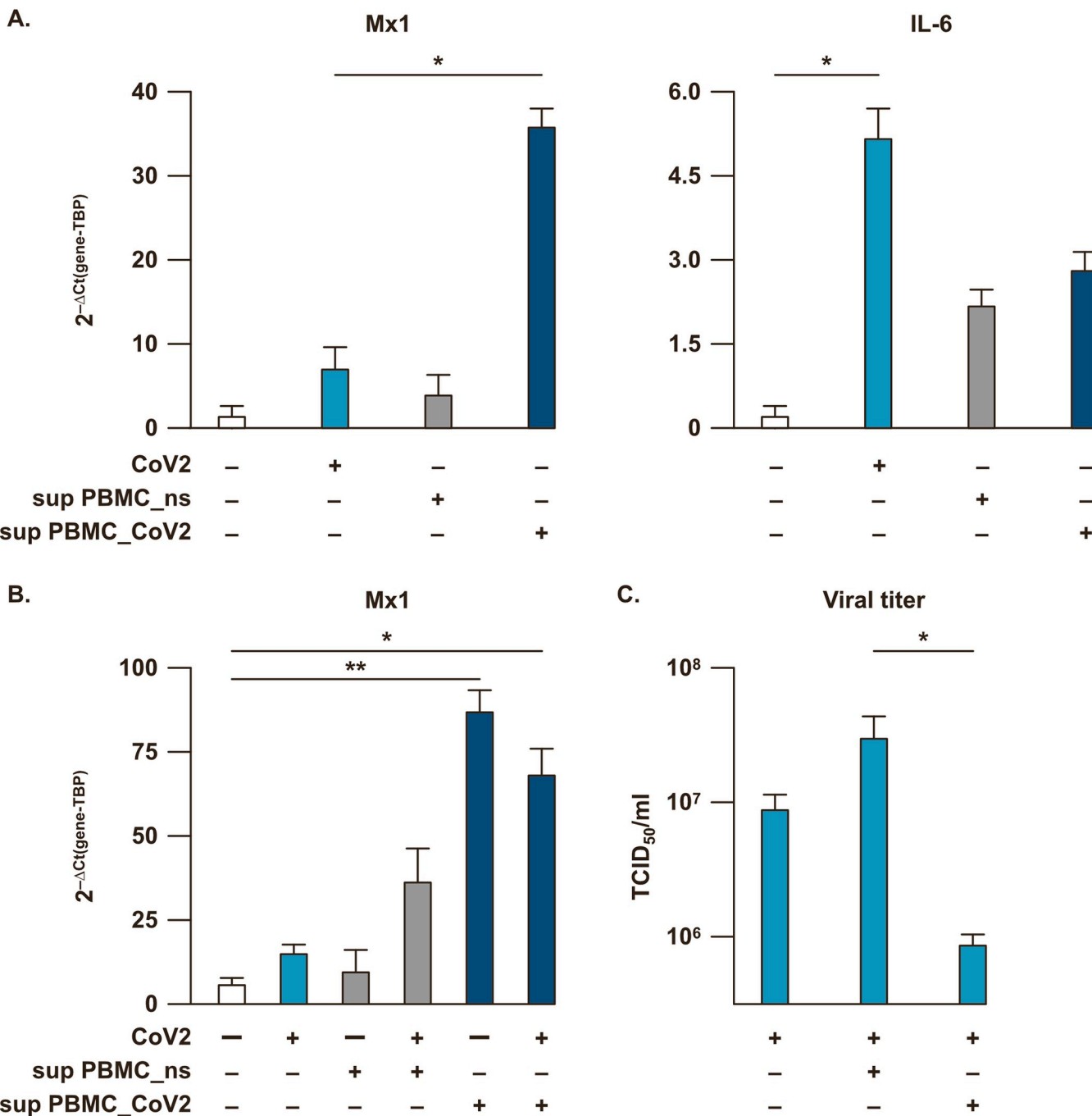

**Fig 3. Type I IFN-induced anti-viral state impacts on SARS-CoV-2 infected Calu-3 lung epithelial cell line.** (**A**) The human lung epithelial cell line Calu-3 was left untreated (not stimulated, ns), infected for 24 hours with SARS-CoV-2 at the back titration dose calculated in peripheral blood mononuclear cells (PBMC) ($10^4$ TCID$_{50}$/ml for 0.02 MOI) or treated with 24 hour-supernatants collected from either ns PBMC cultures (sup PBMC_ns) or from SARS-CoV-2 (MOI = 0.02)-treated PBMC cultures (sup PBMC_CoV2). Relative expression of Mx1 and IL-6 genes was measured by quantitative real time PCR analysis. All quantification data were normalized to TBP level by using the equation $2^{-\Delta Ct}$. (**B**) Calu-3 cell line was infected with SARS-CoV-2 at the back titrated dose calculated in PBMC ($10^4$ TCID$_{50}$/ml for 0.02 MOI) in presence of 24 hour-supernatants collected from either ns PBMC cultures (sup PBMC_ns) or from SARS-CoV-2 (MOI = 0.02)-treated PBMC cultures (sup PBMC_CoV2). Mx1 gene expression was quantified by real time PCR and quantification data normalized to TBP level by using the equation $2^{-\Delta Ct}$. (**C**) Virus titers were evaluated in supernatants from infected Calu-3 cultures described in (**B**) by Endpoint Dilution Assay by using the Reed-Muench formula and reported as TCID$_{50}$/ml. Shown results were mean relative values ± SEM of 3 independent experiments. P-values were depicted as follows: *p≤0.05; ** p≤0.001.

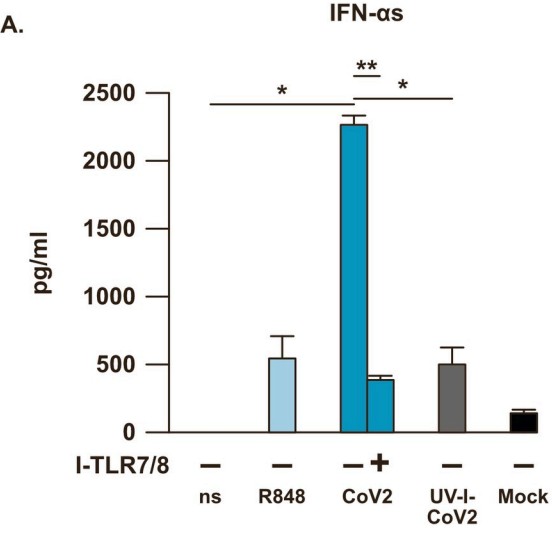

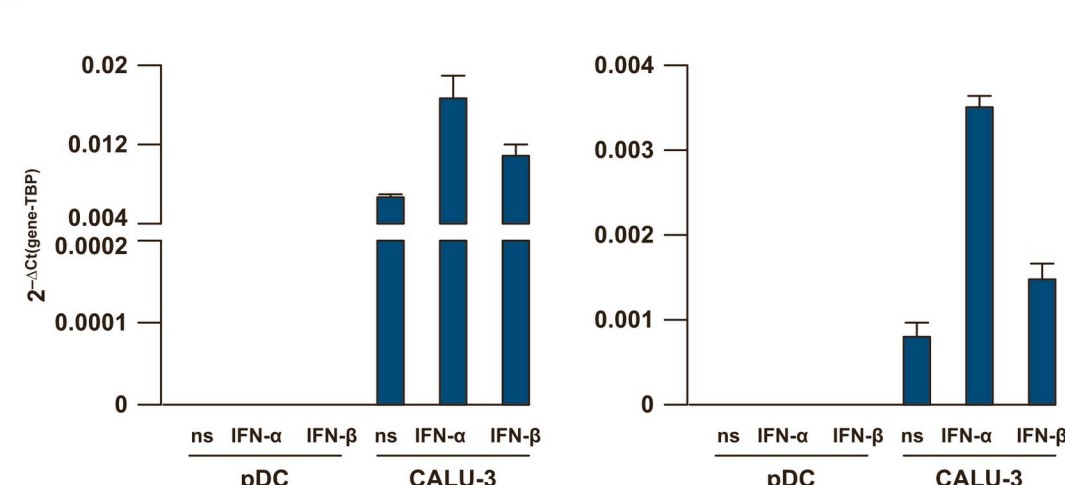

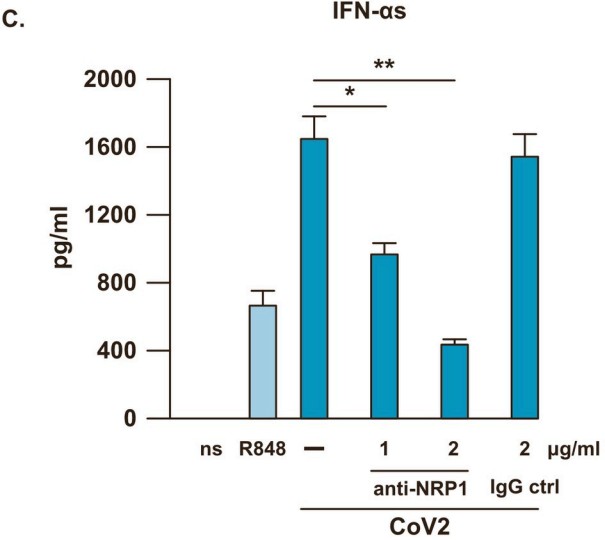

**Fig 4. SARS-CoV-2-induced IFN-α production is dependent on TLR7 and Neuropilin 1 in pDC.** (**A**) Purified plasmacytoid dendritic cells (pDC) were left untreated (not stimulated, ns) or Mock-treated as negative controls, or stimulated for 24 hours with the TLR7/8 agonist R848 (5μM) as positive control, with SARS-CoV-2 (CoV2; MOI = 0.1) in presence or absence of a 30 minute pre-treatment with a specific TLR7/8 inhibitor (I-TLR7/8, 1μM) or with UV-I-SARS-CoV-2 (UV-I-CoV2, MOI = 0.1). IFN-α production was tested by ELISA in collected cell culture supernatants. (**B**) pDC and Calu-3 cell line were stimulated with 1000 U/mL of recombinant IFN-α or IFN-β. Gene expression of ACE-2 and TMPRSS2 was measured by quantitative RT-PCR. (**C**) pDC were treated with CoV2 (MOI = 0.1) in presence or absence of a 30 minute pre-treatment with increasing doses of anti-Neuropilin 1 monoclonal antibody (anti-NRP1, 1 or 2 μg/ml) or anti-human IgG control (IgG ctrl, 2 μg/ml). IFN-α production was tested by ELISA in culture supernatants. Shown results were mean relative values ± SEM of 3 independent experiments. P-values were depicted as follows: $^*p{\leq}0.05$; $^{**}$ $p{\leq}0.001$.

Similarly to pDC, PBMC stimulation by UV-I-SARS-CoV-2 affects both cytokine and chemo-kine production (**S4A and S4B Fig**).

Differently to what occurs in human airway epithelial cells [27] and in Calu-3 cells, SARS-CoV-2 stimulation of pDC did not depend on ACE-2 mediated viral entry or on the trans-membrane serine protease 2 (TMPRSS2), both not expressed by these cells in baseline condi-tion or after stimulation with IFN-αs and IFN-β (**Fig 4B**). Interestingly, *in vitro* stimulation by SARS-CoV-2 of monocytes, which exclusively express ACE-2 but not TMPRSS2 (**S5A Fig**), did not lead to IFN-α or IL-6 production as compared to R848 treatment (**S5C Fig**), even if these cells can be efficiently infected by the virus [28]. The responsiveness of pDC, monocytes and Calu-3 to exogenous IFN-αs and IFN-β was tested by evaluating the induction of Mx1 to verify the *bona fide* of ACE-2 and TMPRSS2 expression data (**S5B Fig**).

We also investigated another recently described SARS-CoV-2 receptor [29], namely Neuro-pilin 1 (NRP1, also denoted as BDCA4), which is also a specific pDC marker [30]. NRP1 block-ing *via* a specific monoclonal antibody significantly impairs IFN-α release in these cells in a dose dependent manner (**Fig 4C**). Thus, our results indicate a possible role for NRP1 as viral entry factor in pDC even in absence of ACE-2 and TMPRSS2 expression.

## SARS-CoV-2 stimulation induces a specific phenotype in human pDC

pDC undergo phenotypical diversification in response to viral infections or single stimuli through environmental plasticity [31]. They can diversify into three stable populations: P1-pDC (PD-L1$^+$CD80$^-$) specialized for type I IFN production, P2-pDC (PD-L1$^+$CD80$^+$) dis-playing both innate and adaptive functions and P3-pDC (PD-L1$^-$CD80$^+$) specifically with adaptive functions [31]. In our experimental setting we monitored pDC phenotype and found a clear-cut increase of the frequency of PD-L1$^+$ P1-pDC, in accordance with the detected high IFN-α release, upon SARS-CoV-2 stimulation (**Fig 5A**). In line with these data, SARS-CoV-2-treated total pDC and P1-pDC display a very high surface expression of PD-L1 (**Fig 5B**). In presence of the TLR-7/8 inhibitor, P1 phenotype upon SARS-CoV-2 stimulation reverted into the more adaptive P2 and P3 populations, similarly to R848-treated cultures (**Fig 5A**), with a strong increase of the surface level of CD80 (**Fig 5B**). Accordingly, along with the maturation process, exemplified by the induction of the co-stimulatory marker CD86 (**Fig 5B**), SARS-CoV-2-treated pDC turn from type I IFN- into TNF-α- and IL-6-producing cells in the pres-ence of the TLR-7/8 inhibitor (**Fig 5C**).

## pDC of asymptomatic and hospitalized COVID-19 subjects display a different phenotype

Having defined *in vitro* the key role of pDC in the induction of a type I IFN-mediated anti-viral state in SARS-CoV-2-treated human PBMC, we then moved to study this cell type *ex vivo* in PBMC collected from individuals with asymptomatic SARS-CoV-2 infection (CP-AS,

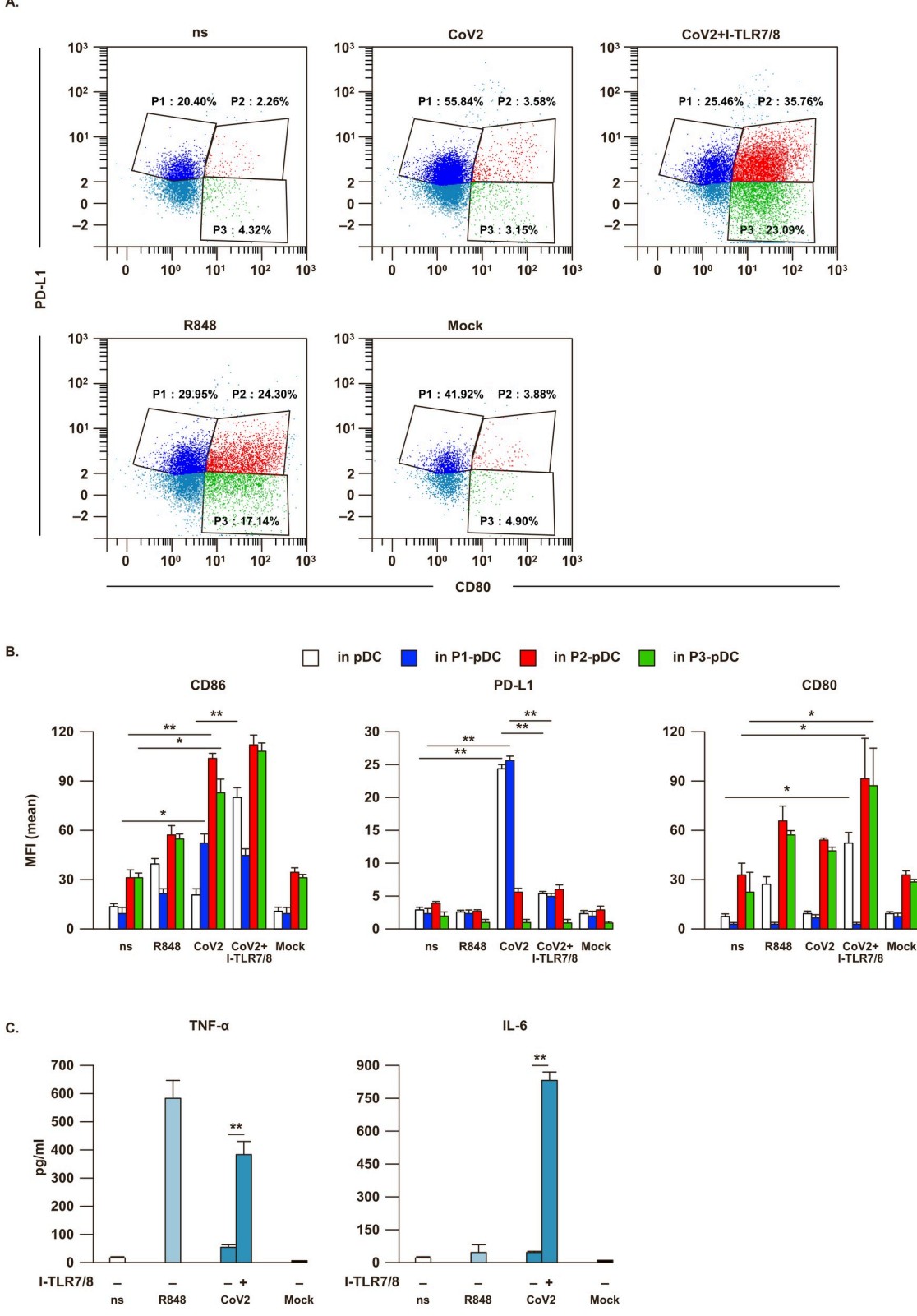

**Fig 5. SARS-CoV-2 stimulation drives TLR7-dependent phenotypical modification in pDC.** Purified plasmacytoid dendritic cells (pDC) were either left untreated (not stimulated, ns) or Mock-treated as negative controls, or stimulated for 24 hours with the TLR7/8 agonist R848 (5μM) as positive control, with SARS-CoV-2 (CoV2; MOI = 0.1) in presence or absence of a 30 minute pre-

treatment with a specific TLR7/8 inhibitor (I-TLR7/8, 1μM). (**A**) pDC were then stained with anti-BDCA4, PD-L1, CD80 and CD86 antibodies. The percentage (%) of pDC sub-populations was evaluated by flow cytometry in live/single BDCA4$^+$ pDC; in particular P1-pDC (PD-L1$^+$CD80$^-$, in blue), P2-pDC (PD-L1$^+$CD80$^+$, in red) and P3-pDC (PD-L1$^-$CD80$^+$, in green). Representative dot plot profile out of 3 different experiments independently conducted is shown. (**B**) Surface expression of CD86, PD-L1 and CD80 was determined as mean fluorescence intensity (MFI) by flow cytometer analysis. (**C**) Production of TNF-α and IL-6 was tested by cytometric bead assay in 24 hour-collected pDC culture supernatants. Shown results were mean relative values ± SEM of 3 independent experiments. P-values were depicted as follows: $^*$p≤0.05; $^{**}$ p≤0.001.

n = 8), in hospitalized COVID-19 patients (CP, n = 6) and in a cohort of healthy donors matched for sex and age (HD, n = 5) (see gating strategy in **S6A Fig** and patient characteristics in **S2** and **S3** **Tables**).

We first monitored whether the degree of COVID-19 severity would match to a differential level of circulating pDC and found a striking difference in the three analyzed groups (**Fig 6A**). The frequency as well as the absolute number of pDC were reduced in asymptomatic SARS-CoV2 infected subjects as compared to HD (**Fig 6A**). According to the observed lymphopenia (**S2 Table**), symptomatic hospitalized patients had a very and consistently low level/depletion of circulating pDC independently of their age and sex (**Fig 6A**).

Further analysis of pDC phenotype in these groups highlighted that CP-AS mainly display a PD-L1$^+$ P1 phenotype; while, on the contrary, in CP mainly PD-L1$^+$CD80$^+$ P2-pDC were observed (**Fig 6B and 6C**). These data were in accordance also to the surface level of both PD-L1 and CD80 markers in pDC expressed in terms of mean fluorescence intensity (MFI) (**Figs 6D** and **S6B**). Importantly, CD86, which was significantly expressed in CP pDC, was further enhanced in CP-AS indicating that asymptomatic infection with SARS-CoV-2 strongly activates pDC (**Figs 6D** and **S6B**).

We then monitored the expression of CXCR3 and CXCR4 on pDC, responsible for their localization to infected peripheral tissues including skin and epithelia, as well as of CCR7 and CD62-L, a chemokine receptor and an integrin both promoting pDC lymph node homing (**Fig 6E and 6F**). SARS-CoV-2 natural infection significantly increased both frequency (**Fig 6E**) and surface expression (**Figs 6F** and **S6C**) of all these chemotactic markers on pDC of COVID-19 patients independently of the degree of disease severity, as compared to the level found in circulating pDC of matched HD, indicating that these cells are committed to migrate to the sites of viral infection.

## Asymptomatic and hospitalized individuals with SARS-CoV-2 infection display a specular anti-viral and pro-inflammatory profile

To investigate if the significant difference in number and activation status observed in pDC of asymptomatic and severe COVID-19 patients would mirror a different anti-viral state in these individuals, we analyzed in *ex vivo* PBMC derived from HD, CP-AS and CP the transcription of the classical ISG Mx1, and found a striking difference in its expression level (**Fig 7A**). RNA from CP-AS PBMC displayed a very high level of this gene as compared to HD, while CP PBMC had a much lower expression, even if significantly higher than HD cells (**Fig 7A**). Interestingly and in accordance with these results, IFN-α level markedly raised in sera of CP-AS as compared to CP and HD (**Fig 7B**).

The pro-inflammatory cytokines IL-6, TNF-α and IL-1β (**Fig 7C**) together with the chemotactic factors CXCL-10, CCL-2 and CXCL-8 (**Fig 7D**) had a specular profile to that observed for IFN-αs. Indeed, they were all found more increased in sera of critical than in asymptomatic COVID-19 subjects (**Fig 7C and 7D**). Interestingly, the serum level of IL-10, a regulatory cytokine with a strong anti-inflammatory potential also known to be induced by type I IFN, was significantly increased also in CP-AS (**Fig 7C**). Similarly, serum CXCL-10, whose transcription

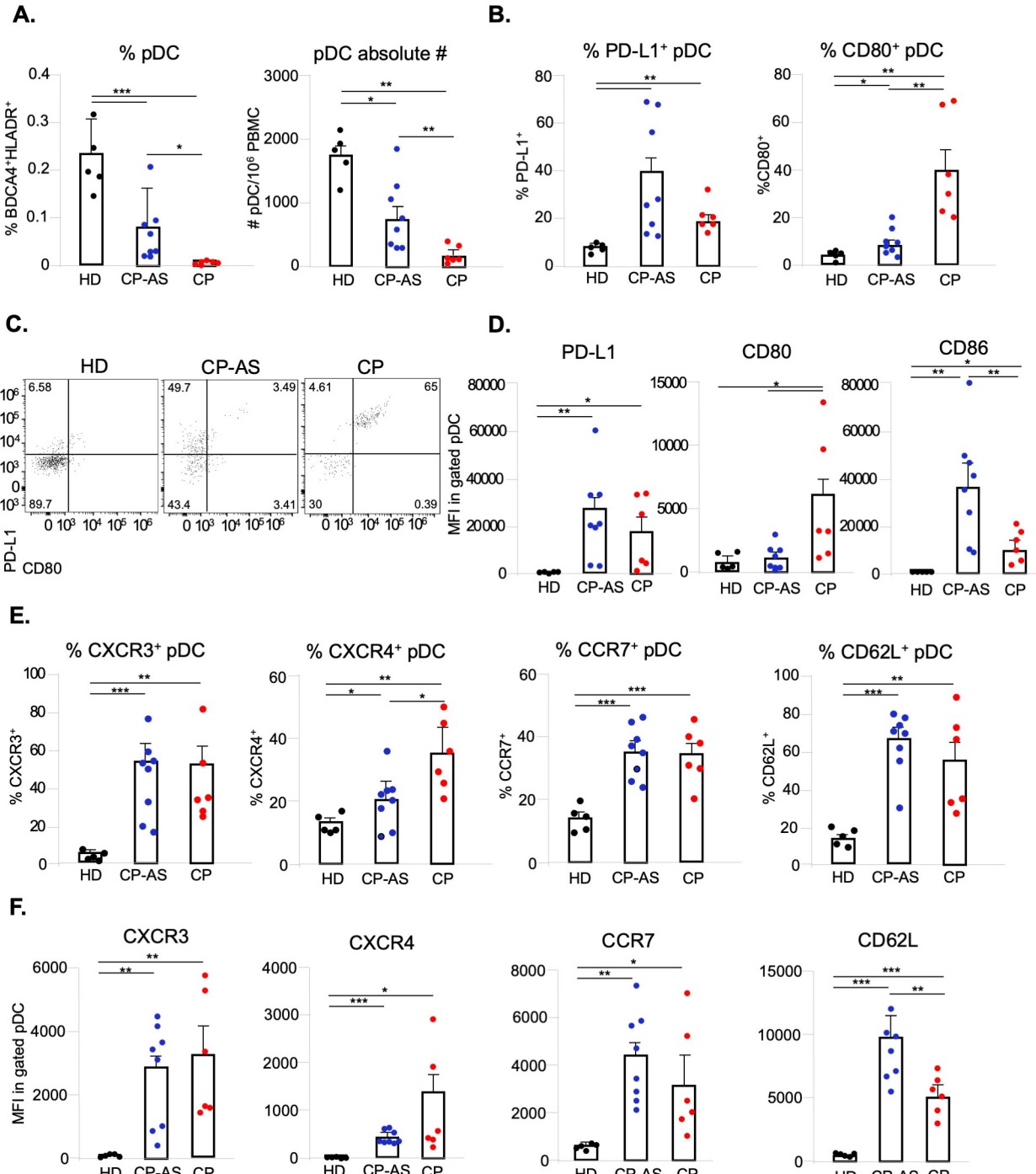

**Fig 6. pDC differently activate and express chemokine receptors in COVID-19 asymptomatic and hospitalized patients.** Freshly isolated peripheral blood mononuclear cells (PBMC) from asymptomatic (CP-AS, n = 8) and hospitalized COVID-19 patients (CP, n = 6) as well as matched healthy donors (HD, n = 5) were stained with a cocktails of antibodies to study by flow cytometry plasmacytoid dendritic cell (pDC) frequency and absolute number (**A**), diversification and activation status (Lineage, CD123, BDCA-4, HLA-DR, PD-L1, CD80 and CD86) (**B, C, D**), as well as expression of chemokine receptors (CXCR4, CXCR3, CCR7 and CD62-L) (**E, F**). The percentage (%) of shown pDC sub-populations (**B, C, E**) was evaluated in live/single Lineage$^-$CD123$^+$BDCA4$^+$HLADR$^+$ gated pDC and depicted for each studied patients or HD together with mean ± SEM values. Surface expression of PD-L1, CD80, CD86 (**D**) and CXCR4, CXCR3, CD62-L, CCR7 (**F**), was determined as mean fluorescence intensity (MFI) and shown results were mean relative values ± SEM of analyzed patients or HD. P-values were depicted as follows: $^*$p≤0.05; $^{**}$ p≤0.001; $^{***}$ p≤0.0001.

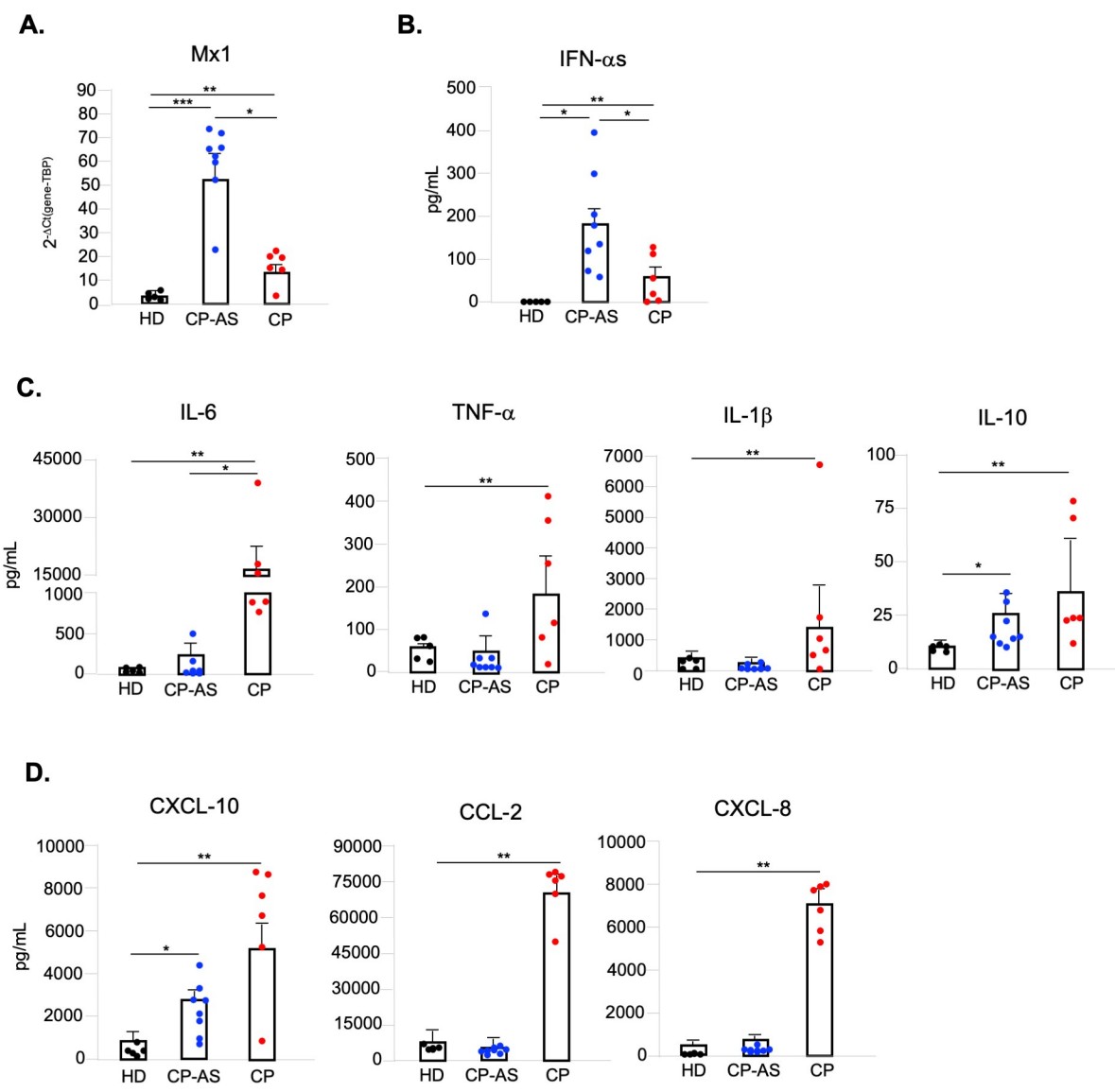

**Fig 7. COVID-19 asymptomatic and hospitalized patients display a specular anti-viral and pro-inflammatory profile.** Peripheral blood mononuclear cells (PBMC) and sera were collected from asymptomatic (CP-AS, n = 8) and hospitalized COVID-19 patients (CP, n = 6) as well as matched healthy donors (HD, n = 5). (**A**) Relative expression of Mx1 gene was measured by quantitative real time PCR analysis and normalized to TBP level by using the equation $2^{-\Delta Ct}$ in total RNA isolated from *ex vivo* PBMC. (**B**) Production of IFN-αs was tested in serum samples by a specific ELISA kit. Production of cytokines (IL-6, TNF-α, IL-1β, IL-10) (**C**) and chemokines (CXCL-10, CCL-2, CXCL-8) (**D**) was tested by multiparametric cytokine bead arrays in collected serum samples. Shown results were mean relative values ± SEM of analyzed patients or HD. P-values were depicted as follows: *p≤0.05; ** p≤0.001; *** p≤0.0001.

is mainly regulated by type II IFN-γ-mediated signaling but in part also by type I IFNs, was enhanced not only in CP but partly also in CP-AS (**Fig 7D**).

## Discussion

Early phases of anti-viral responses mainly consist of two interconnected pathways: first, the engagement of cellular anti-viral defenses, which are mediated by the transcriptional induction of type I and III IFNs with subsequent up-regulation of ISGs [32] and, second, the recruitment and coordination of specific subset of leukocytes primarily arranged by chemokine secretion

[33]. In a variety of infection models, including *in vitro* infection of permissive lung epithelium cell lines and primary bronchial epithelial cells as well as *ex vivo* samples derived from COVID-19 patients and animals, it was shown that, despite viral replication, a waning host immune response was induced with an altered induction of type I and III IFNs associated to a high release of chemokines [12,22]. This reduced expression of type I, II and III IFNs is relevant for SARS-CoV-2, which replicates more efficiently than SARS-Co-V [34]. In addition, as for other coronaviruses that have evolved several escaping strategies to inhibit IFN anti-viral action [35,36], different SARS-CoV-2-encoded proteins were shown to limit type I IFN functions or pathways [37]. In particular, nsp13, nsp14, nsp15, ORF6, ORF8, ORF3b and nucleocapsid proteins act as competent suppressors of IFN-β; while nsp1, nsp12, nsp13, nsp15 and the M protein are also potent inhibitors of the MAVS pathway [38–40].

Accordingly, *in vivo* data from severely ill SARS-CoV-2-infected patients confirmed an impaired type I IFN response accompanied by elevated pro-inflammatory cytokine levels in sera [7,22], thus identifying exacerbated and abnormal responses in the innate branch of the immune system as a main driver of major illness [12].

In the present study, we interrogated human PBMC, a mixture of highly specialized immune cells, to understand their contribution to COVID-19 immunopathogenesis and the dynamic interaction between immune system and SARS-CoV-2 virus. We showed that human PBMC, even if not allowing a productive viral replication, respond to stimulation with a SARS-CoV-2 clinical isolate by expressing high levels of type I and III IFNs and a concomitant ISG transcription, as well as of the inflammatory cytokines IL-6, TNF-α and IL-1β.

It is well known that lymphopenia, together with neutrophilia and monocytopenia, is a characteristic of patients with severe COVID-19 [41]. Consistently with these data, one of the main mechanisms associated with disease severity is the recruitment of inflammatory immune cells towards infected lungs and the subsequent hyperinflammatory state or the so-called cytokine releasing syndrome, which includes inflammatory mediators as well as chemokines [6,42]. A dramatic induction of chemotactic factors was found in *post-mortem* lung samples and in plasma or sera of COVID-19 patients at different stages of disease [43]. In particular, levels of CXCL-10, involved in the etiology of various pulmonary conditions and attracting monocytes, NK and CXCR3-expressing Th1 cells [44,45], increased with disease severity, suggesting that it may help in early diagnosis and could serve as a potential predictive marker of disease outcome [42]. Also, the monocyte chemoattractant molecule CCL-2, was found upregulated during the early phase of infection, further increased during late stages of fatal disease [46] and associated with a prolonged duration of intensive care unit stay [47,48]. Patients with fatal COVID-19 also showed significantly higher plasma levels of the neutrophil recruiting CXCL-8 [49] as well as of the T and NK cell-attracting CXCL-9 [50], as compared to severe and/or mild COVID-19 patients [22,46]. In our study, *in vitro* SARS-CoV-2 stimulation of PBMC recapitulates the *in vivo* scenario found in COVID-19 since we observed a strong release of the aforementioned chemokines emerged as crucial in COVID-19 pathogenesis.

Further, we proved that SARS-CoV-2-mediated release of inflammatory cytokines and chemokines heavily relies on the TLR-7/8 signaling and requires stimulation with infectious live SARS-CoV-2 virions, consistently with recently published evidences of SARS-CoV-2 partially productive infection of lymphomononuclear cell subsets [28,51].

A characteristic of severe COVID-19 cases and of *in vitro* infected SARS-CoV-2 permissive respiratory epithelial cells is the highly impaired type I IFN release and response (with no IFN-β and low IFN-α production and activity detected), thus allowing and sustaining viral replication and an exacerbated inflammatory response [7,22]. With our experiments on permissive Calu-3 cells conditioned with supernatants of SARS-CoV-2-stimulated PBMC, we proved, for the first time, that PBMC-derived type I and III IFNs are bioactive and strongly enhanced ISG

transcription in Calu-3 cells, differently to what found in the unconditioned SARS-CoV-2 infected counterpart. Consistently with an enhanced anti-viral state, SARS-CoV-2-stimulated PBMC supernatant inhibits viral replication in Calu-3 cells. Importantly, our results are fully coherent with previous data showing that *in vitro* administration of type I IFN-β [52] and type III IFN-λ [53] restrains and reduces SARS-CoV-2 viral replication and with recent data showing that co-culture of pDC with human primary epithelial cells induces an anti-viral response in these cells and results in the control of SARS-CoV-2 viral replication [54].

Having in mind all these evidences, we investigated which immune cell type is responsible for type I IFN release upon SARS-CoV-2 stimulation. pDC are known to be the principal cell type among immune cells specialized in type I IFN release sensing viral RNA in a TLR7-dependent manner [13]. pDC can sense almost all viruses, including the coronaviruses SARS-CoV and MERS [14–16], and were shown to recognize MERS *via* TLR7 [16]. In our study, pDC isolated from peripheral blood of healthy donors and stimulated *in vitro* with SARS-CoV-2 produced a high amount of TLR7-mediated IFN-αs, with a level comparable to that found in PBMC cultures, even if pDC do not express both the entry factors ACE-2 or TMPRSS2 (our data and [55]). However, CD123[bright] pDC constitutively express on their surface NRP1 (also named BDCA4) that acts as an identification marker for this cell type in the blood [30]. Recently it was shown that co-transfection of ACE-2 and TMPRSS2 together with NRP1 in 293-T embryonic kidney cells markedly enhanced the SARS-CoV-2 infection rate observed in presence of ACE-2 and TMPRSS2 alone by facilitating viral entry [14]. The presence of a polybasic furin-type cleavage site at the S1-S2 junction in the SARS-CoV-2 spike protein, that is absent in SARS-CoV [56], likely exposes a spike C-terminal motif that can bind and activate both NRP1 and NRP2 proteins. Thus, by blocking NRP1 expressed on pDC with a specific monoclonal antibody we observed that SARS-CoV-2-mediated production of IFN-α is dramatically impaired suggesting a possible role for viral entry of this molecule also in cells lacking ACE-2 and TMPRSS2 expression, such as pDC. Similarly to total PBMC, then, production of the anti-viral type I IFN by pDC mostly requires a stimulation with infectious SARS-CoV-2 virus since UV inactivation strongly decreased IFN-α production in our culture settings. Other viruses, such as Dengue virus, behaves in a similar manner than SARS-CoV-2 [57]. A different observation was instead made by Cervantes-Barragan and colleagues [54], who used a SARS-CoV-2 cDNA clone generated by a reverse genetic system [58]. In their settings UV-inactivation of SARS-CoV-2 does not seem to reduce IFN-α production and ISG expression in pDC [54]. The discrepancy between our and their data may rely on the use of the UV inactivated recombinant virus, in Cervantes-Barragan's paper, at a concentration 10-times higher than the one we used in this study performed with the clinical viral isolate (MOI 1 *versus* 0.1). Thus, these extreme experimental conditions may push the *in vitro* system to a level that would be difficultly observed *in vivo* and that may also induce a plateau expression level for ISGs whose transcription is very sensitive and rapidly induced by even very low amounts of type I IFN.

Further, our study showed that stimulation with SARS-CoV-2 regulated pDC phenotype and activation status. Hence, viral sensing mediates the cell surface up-regulation of PD-L1 on more of the 55% of cultured pDC, a phenotype found in pDC specialized in type I IFN production [31]. In presence of a TLR-7/8 inhibitor SARS-CoV-2-treated pDC turn from PD-L1[+]CD80[-] to PD-L1[+]CD80[+] or CD80[+] only expressing cells, two subpopulations known to mediate adaptive functions and T cell interaction [31], as exemplified in our study also by the up-regulation of the costimulatory and maturation-associated marker CD86. Accordingly, pDC in presence of SARS-CoV-2 and TLR7/8 inhibitor do not release type I IFNs and turn into TNF-α- and IL-6-producing cells. Production of the pro-inflammatory cytokine TNF-α, in particular, is known to negatively control in an autocrine/paracrine fashion the production

of IFN-αs in pDC as soon as the maturation process starts [59]. Similarly to what found in SARS-CoV-2-treated cells, the treatment with a TLR7/8 inhibitor blocking type I IFN production enhances TNF-α and IL-6 also in pDC stimulated with another single-stranded RNA virus, Tick-borne encephalitis virus, inducing a similar phenotypic diversification of pDC than that observed with SARS-CoV-2 [60].

While we were finalizing this paper, Onodi and colleagues published similar results showing that SARS-CoV-2, in absence of productive infection, induces pDC phenotype diversification at a level similar to influenza virus [55]. SARS-CoV-2-mediated type I IFN production was then blocked in pDC by hydroxychloroquine [55], in line to what we found in presence of a specific TLR7/8 inhibitor.

SARS-CoV-2 rapidly impairs T cell and DC responses during the acute phase of infection, which could have significant implications for COVID-19 pathogenesis [17]. Moreover, pDC have been extensively studied in patients with severe COVID-19 and were found significantly reduced or absent in the blood of these individuals [17–19]. Importantly, while inflammatory monocytes and CD1c+ conventional DC were present in lung infiltrates of patients with severe COVID-19, CD123hi pDC were depleted in blood but absent in the lungs [20].

While Onodi *et al*. demonstrated *in vitro* that pDC activation by SARS-CoV-2 is dependent on IRAK4- and UNC93B1-mediated signaling pathways by means of patients genetically deficient in genes encoding these factors [55], in this study we deepen the current understanding on pDC contribution and phenotype diversification in SARS-CoV-2 infection by characterizing *ex vivo* blood samples derived from patients at different COVID-19 disease severity. As expected, we observed a profound pDC depletion, in terms of both frequency and absolute numbers, in patients hospitalized with severe COVID-19 as compared to matched HD. Interestingly, then, in a cohort of asymptomatic SARS-CoV-2-infected individuals circulating pDC were already reduced in the peripheral blood as respect to the healthy counterpart but significantly higher than what found in hospitalized COVID-19 patients. In accordance with these data, a recent paper performed a single-cell RNA sequencing of PBMC from moderate or severe COVID-19 and highlighted increased pro-apoptotic pathways in pDC of severely infected patients pointing towards a mechanism that may explain insufficient control of SARS-CoV-2 infection [61]. Nonetheless, we also found that the phenotype was drastically different in pDC derived from severely infected or asymptomatic subjects. In the latter group, pDC mainly expressed PD-L1, while cells derived from severe COVID-19 were consistently represented by PD-L1+CD80+ phenotype. Importantly, the analysis of costimulatory marker CD86, that was significantly expressed in pDC from critically ill patients and further enhanced in cells of asymptomatic subjects, indicated that pDC are strongly activated during asymptomatic infection.

We also observed that the expression of chemokine receptors responsible for pDC localization to infected peripheral tissues, such as CXCR3 and CXCR4, or for their lymph node homing, as CCR7 and CD62-L, was induced by SARS-CoV-2 natural infection both in terms of frequency and surface expression levels in cells of both asymptomatic or severe COVID-19 patients, possibly indicating that these cells are committed to migrate to the sites of viral infection independently of the degree of disease severity.

Hence, having found dramatic differences in phenotype of pDC from asymptomatic or severe COVID-19, we also tested if these data matched to the induced cytokine profile in the different disease courses. Most interestingly, PBMC isolated *ex vivo* from HD, asymptomatic or severely infected subjects indicated that anti-viral ISG expression is massively induced only in asymptomatic individuals. These data matched to the high amount of IFN-αs present in sera and with the PD-L1+ P1 phenotype, specialized in type I IFN production, found in pDC of asymptomatic SARS-CoV-2 infected subjects. These results are in line with recent findings

proving the high frequency of neutrophils and monocytes expressing high levels of ISG only in mild COVID-19, and not in the severe form [12]. These ISG are likely up-regulated in these cell types in response to the high amount of circulating IFN-αs released by PD-L1$^+$ pDC.

Lastly, our study also demonstrated that a specular profile of anti-viral and inflammatory cytokines and chemokines exists between sera samples derived from asymptomatic or severe SARS-CoV-2 infection. We found that severe COVID-19 patients displayed, as extensively already reported [50,62,63], high level of circulating inflammatory cytokines, including IL-6, TNF-α, IL-1β and chemokines such as CXCL-10, CCL-2 and CXCL-8 contributing to the diffuse inflammatory status and the so-called cytokine storm. Conversely, we demonstrated that subjects with asymptomatic infection showed high level of IFN-αs and of the immune-regulatory IFN-inducible IL-10 known to dampen ongoing inflammatory responses.

In conclusion, in our study by using an *in vitro* human PBMC-based experimental model we recapitulated the *in vivo* scenario found in early SARS-CoV-2 infection and assessed the importance of pDC response and pDC-induced type I IFN in the regulation of the anti-viral state in asymptomatic and severe COVID-19 patients. Thus, the PBMC-based experimental setting might represent an optimal tool to study SARS-CoV-2-induced immune responses. Modulating innate antiviral immunity and, in particular, the pDC/type I IFN axis since the early phases of COVID-19 may help to pinpoint novel pharmacological strategies or host-directed therapies that would counter-act the raising of hyper-inflammation and the resulting diffuse damage contributing to a rapid resolution of SARS-CoV-2 infection.

## Materials and methods

### Ethics statement

Istituto Superiore di Sanità Review Board approved the use of blood from HD (AOO-ISS—14/06/2020–0020932) and from asymptomatic SARS-CoV2 infected individuals (IN_COVID, AOO-ISS—22/03/2021–0010979). Policlinico Tor Vergata approved blood withdrawal from hospitalized COVID-19 patients (COVID-SEET, CE#46.20, 18/04/2020).

### Patients

In particular, for this study six hospitalized COVID-19 [CP; 2 females/4 males; median age ± Standard Deviation (SD) 51.5 ± 23.3 yrs.] and eight asymptomatic [CP-AS; 4 females/4 males; 58.5 ± 10.4 yrs.] patients matched to five HD [2 females/3 males; 53 ± 12.5 yrs.] were enrolled and provided written informed consent. Main demographic, clinical and experimental data related to asymptomatic and hospitalized COVID-19 patients are listed in S2 and S3 Tables.

### Isolation and culture of PBMC, pDC and monocytes

PBMC were collected from peripheral blood and isolated and cultured as described [60]. pDC and monocytes were purified from isolated PBMC by magnetic separation by using anti-BDCA4 and anti-CD14 microbeads (Miltenyi biotech) respectively, as previously described [64]. The purity of the recovered cells was greater than 95% as assessed by flow cytometry analysis with anti-BDCA4 (Miltenyi biotech) or anti-CD14 (BD Biosciences) monoclonal antibodies.

### Virus production

Vero E6 (Vero C1008, clone E6-CRL-1586; ATCC) cells were cultured in Dulbecco's Modified Eagle Medium (DMEM) supplemented with non-essential amino acids (NEAA, 1x),

penicillin/streptomycin (P/S, 100 U/mL), HEPES buffer (10 mM) and 10% (v/v) Fetal bovine serum (FBS). The clinical isolates of SARS-CoV-2 hCoV-19/Italy/UniSR1/2020 (GISAID accession ID: EPI_ISL_413489) was isolated and propagated in Vero E6 cells, and viral titer was determined by 50% tissue culture infective dose ($TCID_{50}$) and plaque assay for confirming the obtained titer.

The influenza virus A/Milan/UHSR1/2009 (GenBank: CQ232099.1) strain was propagated in MDCK in medium supplemented with 2 μg/ml TPCK–trypsin.

## Virus titration

SARS-CoV-2 stocks and supernatants of the experimental conditions were titrated using End-point Dilutions Assay (EDA, $TCID_{50}$/mL). Vero E6 cells ($4 \times 10^5$ cells/mL) were seeded into 96 wells plates and infected with base 10 dilutions of collected medium, each condition tested in triplicate. After 1 h of adsorption at 37°C, the cell-free virus was removed, and complete medium was added to cells. After 72 h, cells were observed to evaluate CPE. $TCID_{50}$/mL was calculated by Endpoint Dilution Assay by using the Reed-Muench formula.

Influenza A viral titer was determined using indirect immunofluorescence system (FFU/mL) and plaque assay (PFU/mL).

## Viral inactivation by UV-C

An aliquot (0.170 mL) of both SARS-CoV-2 and Influenza A viral stocks was place in a 24-well plate in ice to counteract irradiation-derived heating of the sample and irradiated with approximately 1.8 mW/cm$^2$ at a work distance of 20 cm for 40 minutes. The viral inactivation was checked using undiluted supernatant in an infection assay [25].

## Calu-3 treatment and infection

Calu-3 (Human lung cancer cell line, ATCC HTB-55) were cultured in MEM supplemented with NEAA (1x), P/S (100 U/mL), Sodium Pyruvate (1 mM), and 10% (v/v) FBS. Calu-3 cells were seeded into 12-wells plates to reach confluence. Before the viral adsorption, the cells were treated with 200 μL of the different PBMC supernatants for 1 hour at 37°C. The viral adsorption was conducted as already described using 5.62 x $10^4$ $TCID_{50}$/mL of hCoV-19/Italy/UniSR1/2020. After then, 500 μL of the different PBMC supernatants were added and the cells were incubated at 37°C for 24 hours.

## Cell stimulation and supernatant collection

PBMC, pDC and monocytes were pre-incubated for 1 hour at 37°C with infectious or UV-inactivated SARS-CoV2 at 0.01, 0.02, 0.04 or 0.1 MOI as well as with UV-inactivated Influenza A virus (MOI = 0.1) and then cultured at 2x$10^6$ cells/ml in RPMI 1640 in presence of P/S (100 U/mL), L-glutamine (2mM) and 10% FBS for 24 or 48 hours. As mock-treatment, cells were stimulated with supernatants from Vero E6 uninfected cells at the same passage as those infected with SARS-CoV-2 at a dilution corresponding to that of MOI 0.1 infected cultures. Cells were also treated with the TLR7 agonist Imiquimod (Imq, 1 or 5μM as specified, Invivogen), the TLR7/8 agonist Resiquimod (R848, 1 or 5μM as specified, Invivogen), the TLR9 ligand type C CpG (CpG-C, 3μg/ml) or with 1000 U/mL of recombinant IFN-α2 and IFN-β (Peprotech). Furthermore, where indicated, PBMC and pDC were also pre-treated for 30 min with 1μM TLR7/8 antagonist, 2087c oligonucleotide (Miltenyi biotech) or with a control oligonucleotide sequence (ODN ctrl, 5′-tcctgcaggttaagt-3′, 1μM) [65] prior to stimulation with

SARS-CoV2. Sheep anti-human Neuropilin-1 Antibody and its control Normal Sheep IgG Control (R&D Systems) were used at 1 or 2μg/ml, as specified.

After 24 or 48 hours, cell culture supernatants were harvested and treated for 30 minutes at 56˚C, then stored at -80˚C for later use. SARS-CoV2 inactivation was tested by back titration for each experiment (see above for details).

## Detection of cytokines and chemokines in culture supernatants

Release of IFN-αs was measured by a specific ELISA kit (PBL assay science). Production of cytokines (IL-6, TNF-α, IL-1β, IL-10 and IL-12p70) and chemokines (CXCL-10, CCL-2, CXCL-9, CXCL-8 and CCL-5) was quantified by specific cytometric bead arrays (BD Biosciences) on a FACS Canto (BD Biosciences) and analyzed by FCAP array software (BD Biosciences).

## Flow cytometry analyses

Monoclonal antibodies anti-Lineage cocktail (Lin), PD-L1, CD80, CD86, HLA-DR, CD123, CXCR-3, CXCR-4, CD62-L and CCR7 as well as IgG1 or IgG2a isotype controls were purchased from BD Biosciences, while BDCA4 from Miltenyi Biotech. To establish cell viability and exclude dead cells from flow cytometry analyses, Fixable Viability Dye (FvDye, eBioscience) was always included in antibody cocktails. In the mixed cell population of PBMC, pDC were considered as those cells in live/single FvDye⁻Lineage⁻CD123⁺BDCA4⁺HLADR⁺ gate (S3A Fig). Cells ($1.5x10^6$ for PBMC or $5x10^4$ for isolated pDC) were incubated with monoclonal antibodies at 4˚C for 30 min and then fixed with 4% paraformaldehyde before analysis on a Cytoflex cytometer (Beckman Coulter). Data were analyzed by Flow Jo software v.10.7 (BD Biosciences). Expression of analyzed cell surface molecules was evaluated using the median fluorescence intensity (MFI). Only viable and single cells were considered for further analysis.

## RNA isolation and quantitative real time PCR analysis

Total RNA was isolated by Trizol Reagent (Invitrogen, Thermo Fischer Scientific), quantified using a Nanodrop2000 spectrophotometer and quality assessed with an established cut-off of ~1.8 for 260/280 absorbance ratio. Reverse-transcription was conducted by Vilo reverse transcriptase kit (Invitrogen, Thermo Fischer Scientific).

Expression of genes encoding Mx1, IFN-αs, IFN-β, IFN-λ1, IL-6, ACE-2 and TMPRSS2 was measured by quantitative real time PCR (q-PCR) using the appropriate TaqMan assay and TaqMan Universal Master Mix II (Applied Biosystems, Thermo Fisher Scientific) on a ViiA7 Instrument (Applied Biosystems, Thermo Fisher Scientific).

The housekeeping gene TATA-box-binding protein (TBP) was used as normalizer. Real time reactions were run at least in duplicates. Sample values for each mRNA were normalized to the selected housekeeping gene using the formula $2^{-\Delta Ct}$.

## Statistical analysis

Statistical analysis was performed using One-way Repeated-Measures ANOVA when three or more stimulation conditions were compared. A two-tailed paired Student's t-test was used when only two stimulation conditions were compared. Results were shown as mean values ± SEM. P ≤0.05 was considered statistically significant. In the figures, star scale was assigned as follows: * = p ≤ 0.05; ** = p ≤ 0.01; *** = p ≤ 0.001.

## Supporting information

**S1 Fig. Human PBMC are not permissive to SARS-CoV-2 infection.** Virus titers were evaluated in viral inoculum (T0) as well as in peripheral blood mononuclear cell (PBMC) supernatants after 24 (T1) and 48 hours (T2) post SARS-CoV-2 infection in cultures treated at a multiplicity of infection (MOI) of 0.01, 0.02, 0.04 and 0.1. Shown results were calculated by Endpoint Dilution Assay by using the Reed-Muench formula and reported as $TCID_{50}$/ml and derived from 3 experiments separately conducted.
(TIFF)

**S2 Fig. Positive and negative controls of TLR7/8 inhibition in peripheral blood mononuclear cells (PBMC) and plasmacytoid dendritic cells (pDC).** (**A**) PBMC were left untreated (not stimulated, ns) or stimulated for 24 hours with the TLR7 ligand Imiquimod (Imq, at 1 and 5μM), the TLR7/8 agonist Resiquimod (R848, at 1 and 5μM), the TLR9 ligand type C CpG (CpG-C, 3μg/ml), SARS-CoV-2 (CoV2) at a multiplicity of infection (MOI) of 0.04 (1 cell, 0.04 TCID50 of virus) and UV-inactivated influenza A (UV-I-Flu-A, 0.1 MOI) in presence or absence of a 30 minute pre-treatment with either a specific TLR7/8 inhibitor ODN-2087 (I-TLR7/8, 1μM) or an unrelated ODN control (ODN-ctrl, 1μM). Production of IFN-αs and IL-6 was tested in collected culture supernatants. (**B**) pDC were left untreated (ns) or stimulated for 24 hours with R848 (1 and 5μM), CpG-C (3μg/ml), CoV2 (0.1 MOI) and UV-Flu-A (0.1 MOI) in presence or absence of a 30 minute pre-treatment with either I-TLR7/8 (1μM) or with an unrelated ODN-ctrl (1μM). Production of IFN-αs, IL-6 and TNF-α was tested in collected culture supernatants. Shown results were mean relative values ± SEM of 3 independent experiments.
(TIFF)

**S3 Fig. Kinetic of cytokine and chemokine expression in PBMC stimulated by SARS-CoV-2.** Peripheral blood mononuclear cells (PBMC) were left untreated (not stimulated, ns) or stimulated for 24 or 48 hours with SARS-CoV-2 (MOI = 0.04 and MOI = 0.1). (**A, B**) Production of IL-6 (**A**), and CXCL-10 and CXCL-9 (**B**) was tested by multiparametric cytokine bead arrays in collected cell culture supernatants. Shown results were mean relative values ± SEM of 3 independent experiments. P-values were calculated by two-tailed Students' t-test and were depicted as follows: $^*p \leq 0.05$; $^{**}p \leq 0.01$.
(TIFF)

**S4 Fig. SARS-CoV-2-mediated cytokine and chemokine production is not induced upon PBMC stimulation with UV-inactivated virus.** Peripheral blood mononuclear cells (PBMC) were left untreated (not stimulated, ns) or stimulated for 24 hours with infectious or UV-inactivated (UV-I) SARS-CoV-2 (CoV2, MOI = 0.04). Production of cytokines (IL-6 and TNF-α) (**A**) and chemokines (CXCL-10, CXCL-9, CCL-2, CXCL-8 and CCL-5) (**B**) was tested by multiparametric cytokine bead arrays in collected cell culture supernatants. Shown results were mean relative values ± SEM of 3 independent experiments. P-values were depicted as follows: $^*p \leq 0.05$; $^{**}$ $p \leq 0.001$.
(TIFF)

**S5 Fig. Analysis of SARS-CoV-2 entry factors and cytokine production in isolated CD14+ monocytes.** Monocytes, plasmacytoid dendritic cells (pDC) and Calu-3 cell line were left not stimulated (ns) or stimulated with 1000 U/mL of recombinant IFN-α or IFN-β. (**A**) Gene expression of ACE-2 and TMPRSS2 was measured by quantitative RT-PCR in monocytes and Calu-3 cells. (**B**) Mx1 was measured by quantitative RT-PCR on monocytes, pDC and Calu-3 cells. (**C**) Monocytes were treated for 24 hours with SARS-CoV-2 (MOI = 0.04 and MOI = 0.1).

Production of IFN-αs and IL-6 was tested by specific ELISA in collected cell culture supernatants. Shown results were mean relative values ± SEM of 3 independent experiments.
(TIFF)

**S6 Fig. Representative gating strategy for pDC phenotypical analysis in COVID-19 patients.** Freshly isolated peripheral blood mononuclear cells (PBMC) from asymptomatic (CP-AS, n = 8) and hospitalized COVID-19 patients (CP, n = 6) and matched healthy donors (HD, n = 5) were stained with a well-established antibody cocktails to study by flow cytometry plasmacytoid dendritic cell (pDC) diversification and activation status as well as expression of chemokine receptors. (**A**) The percentage (%) of total pDC was evaluated in Fixable viability dye (FvDye)$^-$ live Lineage-CD123$^+$BDCA4$^+$ gated cells. In pDC we also evaluated the % of PD-L1 or CD80 expressing cells. (**B, C**) Surface expression of PD-L1, CD80, CD86 (**B**) and CXCR4, CXCR3, CCR7, CD62-L (**C**) was determined as mean fluorescence intensity (MFI) in CP or HD gated pDC. Shown dot plots and histograms are representative of all analyzed CP and HD.
(TIFF)

**S1 Table. Evaluation of cell death in human PBMC cultures treated with live SARS-CoV2.**
(TIFF)

**S2 Table. Main demographic and clinical characteristics of hospitalized COVID-19 patients.**
(TIFF)

**S3 Table. Main demographic and clinical characteristics of asymptomatic COVID-19 patients.**
(TIFF)

## Author Contributions

**Conceptualization:** Martina Severa, Marilena P. Etna, Eliana M. Coccia.

**Data curation:** Martina Severa, Marco Iannetta, Stefano Balducci, Nicola Clementi.

**Formal analysis:** Nicasio Mancini, Massimo Clementi, Massimo Andreoni, Paola Stefanelli.

**Funding acquisition:** Martina Severa, Luisa Barzon, Eliana M. Coccia.

**Investigation:** Martina Severa, Roberta A. Diotti, Marilena P. Etna, Fabiana Rizzo, Stefano Fiore, Daniela Ricci, Alessandro Sinigaglia, Alessandra Lodi, Elena Criscuolo.

**Methodology:** Roberta A. Diotti, Fabiana Rizzo, Stefano Fiore, Daniela Ricci, Alessandro Sinigaglia, Alessandra Lodi, Elena Criscuolo, Stefano Balducci.

**Project administration:** Martina Severa, Marilena P. Etna, Eliana M. Coccia.

**Supervision:** Luisa Barzon, Paola Stefanelli, Nicola Clementi, Eliana M. Coccia.

**Writing – original draft:** Martina Severa, Marilena P. Etna, Luisa Barzon, Nicola Clementi, Eliana M. Coccia.

**Writing – review & editing:** Martina Severa, Eliana M. Coccia.

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
