## [Decision Letter · Decision Letter 0]

2 May 2021

Dear Dr Coccia,

Thank you very much for submitting your manuscript "Differential plasmacytoid dendritic cell phenotype and type I IFN response in asymptomatic and severe COVID-19 infection" for consideration at PLOS Pathogens. As with all papers reviewed by the journal, your manuscript was reviewed by members of the editorial board and by several independent reviewers. In light of the reviews (below this email), we would like to invite the resubmission of a significantly-revised version that takes into account the reviewers' comments.

In particular, Reviewer #2 made comments about a need for additional controls and enhanced discussion of prior papers on pDCs and COVID-19. Reviewer #1 also made some experimental suggestions to consider.

We cannot make any decision about publication until we have seen the revised manuscript and your response to the reviewers' comments. Your revised manuscript is also likely to be sent to reviewers for further evaluation.

Sincerely,

Michael S. Diamond

Section Editor

PLOS Pathogens

Kasturi Haldar

Editor-in-Chief

PLOS Pathogens

orcid.org/0000-0001-5065-158X

Michael Malim

Editor-in-Chief

PLOS Pathogens

orcid.org/0000-0002-7699-2064

Reviewer's Responses to Questions

**Part I - Summary**

Reviewer #1: The study by Martina Severa and colleagues shows that in vitro plasmacytoid DCs (pDCs) sense Covid-19 via a TLR7-dependent mechanism, leading to Type I IFN secretion. This Type I IFN will protect lung epithelial cells from viral replication and from releasing high levels of inflammatory cytokines, such as IL-6. Furthermore, pDCs stimulated with Covid-19 upregulate PD-L1 but not CD80, suggesting that they specialize in IFN-I production rather than in antigen presentation/costimulation. The authors also show that pDC percentages/numbers are reduced in Covid-19 patients in vivo, both in asymptomatic as well as in severe cases. However, in asymptomatic patients pDCs display a prevalent PD-L1+CD80- phenotype, while severe patients exhibit a PD-L1+CD80+ phenotype. Accordingly, asymptomatic patients display high levels of IFN-I induced genes and, in general, more IFN-a in sera, while severe patients have more inflammatory cytokines in sera. The authors conclude that Type I IFN is protective in Covid-19 infection, in agreement with previous studies showing that genetic defects in the IFN-I pathway predispose to severe Covid-19 pathology.

Overall, this is a well-conducted and informative study.

Reviewer #2: In this article entitled “Differential plasmacytoid dendritic cell phenotype and type I IFN response in asymptomatic and severe COVID-19 infection”, authors report interesting findings on the cellular source of type I IFN following Sars-CoV-2 challenge. They show a major role of pDC in this process. Interestingly, they were also able to analyse patient samples in the context of asymptomatic, or severe COVID-19. However, we have several concerns about the manuscript at this stage.

**Part II – Major Issues: Key Experiments Required for Acceptance**

Reviewer #1: There are a couple of points that should be experimentally addressed:

a) One of the most interesting aspects of the study is that UV-inactivated Covid-19 does not stimulate production of IFN-a by PBMCs. This begs the question what about isolated pDCs? If purified pDCs also do not respond to UV inactivated Covid-19 (differently from what observed for influenza virus and HSV), it would maybe suggest the requirement for a nonstructural viral protein for induction of IFN-I. A similar phenomenon was observed for Dengue type 2 virus (Pichyangkul et. Al. JI. 2003).

b) pDCs express neuropilin1 (BDCA-4), which was recently implicated in cell-entry of Covid-19. Does blockade of neuropilin1 reduce IFN-I production by pDCs?

Reviewer #2: 1. The introduction should be more comprehensive about published work on pDC in COVID-19. This is the main topic of the study. Several references are cited by the authors, but the associated work is not presented in the introduction: in particular Refs 39, 45, and 47-49. Authors should also introduce pDC response to other coronaviruses, which forms important background information: refs 42 and 44. On the contrary, other parts of the introduction appear less relevant to the present study, and could be moved to discussion.

2. In Figures 2, 3, and 6, important controls are missing:

The TLR7/8 inhibitor should be used in the presence of R848 in order to show efficient inhibition (positive control).

Control inhibitor (negative control: non-effective compound of similar chemical nature), should be used in comparison to the effective inhibitor for each viral titer.

The authors introduce Mock as medium only (Fig legend: “Mock medium only »). This is not the usual meaning for Mock. If the condition is Medium only, it should be marked as such on the figure and its legend.

**Part III – Minor Issues: Editorial and Data Presentation Modifications**

Reviewer #1: None

Reviewer #2: 1. Figure 1 presents descriptive negative results and should be shown as supplementary figure, unless the authors provide mechanistic insight.

2. Figure 4 legend mentions « in presence or absence of a 30 minute pre-treatment with a specific TLR7/8 inhibitor (I-TLR7/8, 1μM) ». However, this is not visible on the actual figure. There is no culture condition with the inhibitor.

3. Results presented in figure 5 are completely expected given the antiviral functions of type I interferons. It could be moved to supplementary.

4. In figure 6D, it is very surprising that the inhibitor induces TNF and IL-6 production by pDC. Could authors coment on that ? Is the figure just mislabled ?

5. In Figure 8 : authors should depict individual patient/donnor results in each graph (as in 8A), instead of histograms.

6. In Figure 8, IL-10 levels are actually very low in all conditions. Are these levels and differences biologically meaningful ? What is the sensitivity limit of the assay ? (should be shown on the graph)

PLOS authors have the option to publish the peer review history of their article (what does this mean?). If published, this will include your full peer review and any attached files.

Reviewer #1: No

Reviewer #2: No
---

## [Decision Letter · Decision Letter 1]

9 Aug 2021

Dear Dr Coccia,

We are pleased to inform you that your manuscript 'Differential plasmacytoid dendritic cell phenotype and type I Interferon response in asymptomatic and severe COVID-19 infection.' has been provisionally accepted for publication in PLOS Pathogens.

Best regards,

Michael S. Diamond

Section Editor

PLOS Pathogens

Michael Diamond

Section Editor

PLOS Pathogens

Kasturi Haldar

Editor-in-Chief

PLOS Pathogens

orcid.org/0000-0001-5065-158X

Michael Malim

Editor-in-Chief

PLOS Pathogens

orcid.org/0000-0002-7699-2064

Reviewer Comments (if any, and for reference):

Reviewer's Responses to Questions

**Part I - Summary**

Reviewer #1: The authors have addressed my main concerns.

Reviewer #2: Manuscript has been adequately revised. Additional experiments were performed and important controls were added. Following the authors reply, we would agree to leave results of old Fig 5 presented in New Fig 3.

**Part II – Major Issues: Key Experiments Required for Acceptance**

Reviewer #1: No major issues were identified.

Reviewer #2: (No Response)

**Part III – Minor Issues: Editorial and Data Presentation Modifications**

Reviewer #1: No minor issues exist at this point.

Reviewer #2: (No Response)

PLOS authors have the option to publish the peer review history of their article (what does this mean?). If published, this will include your full peer review and any attached files.

Reviewer #1: No

Reviewer #2: No

---

## [Editor Report · Acceptance letter]

12 Aug 2021

Dear Dr Coccia,

We are delighted to inform you that your manuscript, "Differential plasmacytoid dendritic cell phenotype and type I Interferon response in asymptomatic and severe COVID-19 infection.," has been formally accepted for publication in PLOS Pathogens.

Best regards,

Kasturi Haldar

Editor-in-Chief

PLOS Pathogens

orcid.org/0000-0001-5065-158X

Michael Malim

Editor-in-Chief

PLOS Pathogens

orcid.org/0000-0002-7699-2064